# West Nile Virus in the State of Ceará, Northeast Brazil

**DOI:** 10.3390/microorganisms9081699

**Published:** 2021-08-10

**Authors:** Flávia Löwen Levy Chalhoub, Eudson Maia de Queiroz-Júnior, Bruna Holanda Duarte, Marcos Eielson Pinheiro de Sá, Pedro Cerqueira Lima, Ailton Carneiro de Oliveira, Lívia Medeiros Neves Casseb, Liliane Leal das Chagas, Hamilton Antônio de Oliveira Monteiro, Maycon Sebastião Alberto Santos Neves, Cyro Facundo Chaves, Paulo Jean da Silva Moura, Aline Machado Rapello do Nascimento, Rodrigo Giesbrecht Pinheiro, Antonio Roberio Soares Vieira, Francisco Bergson Pinheiro Moura, Luiz Osvaldo Rodrigues da Silva, Kiliana Nogueira Farias da Escóssia, Lindenberg Caranha de Sousa, Izabel Leticia Cavalcante Ramalho, Antônio Williams Lopes da Silva, Leda Maria Simōes Mello, Fábio Felix de Souza, Francisco das Chagas Almeida, Raí dos Santos Rodrigues, Diego do Vale Chagas, Anielly Ferreira-de-Brito, Karina Ribeiro Leite Jardim Cavalcante, Maria Angélica Monteiro de Mello Mares-Guia, Vinícius Martins Guerra Campos, Nieli Rodrigues da Costa Faria, Marcelo Adriano da Cunha e Silva Vieira, Marcos Cesar Lima de Mendonça, Nayara Camila Amorim de Alvarenga Pivisan, Jarier de Oliveira Moreno, Maria Aldessandra Diniz Vieira, Ricristhi Gonçalves de Aguiar Gomes, Fernanda Montenegro de Carvalho Araújo, Pedro Henrique de Oliveira Passos, Daniel Garkauskas Ramos, Alessandro Pecego Martins Romano, Lívia Carício Martins, Ricardo Lourenço-de-Oliveira, Ana Maria Bispo de Filippis, Alex Pauvolid-Corrêa

**Affiliations:** 1Laboratório de Flavivírus, Fundação Oswaldo Cruz (Fiocruz), Ministério da Saúde (MS), Rio de Janeiro, RJ 21040-900, Brazil; flaviallevy@yahoo.com.br (F.L.L.C.); angelicamguia@gmail.com (M.A.M.d.M.M.-G.); viniciusgcampos@hotmail.com (V.M.G.C.); nielircf@gmail.com (N.R.d.C.F.); marcosclm@ioc.fiocruz.br (M.C.L.d.M.); ana.bispo@ioc.fiocruz.br (A.M.B.d.F.); 2Agência de Defesa Agropecuária do Estado do Ceará (ADAGRI), Fortaleza, CE 60811-520, Brazil; eudson.maia@adagri.ce.gov.br (E.M.d.Q.-J.); williams.lopes@adagri.ce.gov.br (A.W.L.d.S.); jarier.oliveira@adagri.ce.gov.br (J.d.O.M.); 3Secretaria Estadual de Saúde do Estado do Ceará (SES-CE), Fortaleza, CE 60060-440, Brazil; bruna.duarte@ufersa.edu.br (B.H.D.); berim.lg@gmail.com (A.R.S.V.); bergsonpmoura@gmail.com (F.B.P.M.); luizuva@gmail.com (L.O.R.d.S.); kiliana.escossia@saude.ce.gov.br (K.N.F.d.E.); insect.berg@gmail.com (L.C.d.S.); nayarapivisan@gmail.com (N.C.A.d.A.P.); ricristhi@gmail.com (R.G.d.A.G.); 4Departamento de Serviços Técnicos, Secretaria de Defesa Agropecuária, Ministério da Agricultura Pecuária e Abastecimento (MAPA), Brasília, DF 70043-900, Brazil; marcos.sa@agricultura.gov.br; 5Fundação BioBrasil, Ituberá, BA 45435-000, Brazil; pedroclima@gmail.com; 6Centro Nacional de Pesquisa para Conservação das Aves Silvestres (CEMAVE), Instituto Chico Mendes de Conservação da Biodiversidade (ICMBio), Ministério do Meio Ambiente (MMA), Cabedelo, PB 58108-012, Brazil; ailtonoliveiraster@gmail.com; 7Seção de Arbovirologia e Febres Hemorrágicas, Instituto Evandro Chagas (IEC), MS, Ananindeua, PA 67030-000, Brazil; liviacasseb@iec.gov.br (L.M.N.C.); lilianechagas@iec.gov.br (L.L.d.C.); hamiltonmonteiro@iec.gov.br (H.A.d.O.M.); liviamartins@iec.gov.br (L.C.M.); 8Laboratório de Mosquitos Transmissores de Hematozoários, Fiocruz, MS, Rio de Janeiro, RJ 21040-900, Brazil; mayconsn@ioc.fiocruz.br (M.S.A.S.N.); aniellya@ioc.fiocruz.br (A.F.-d.-B.); lourenco@ioc.fiocruz.br (R.L.-d.-O.); 9Autonomous Veterinarian, Boa Viagem, CE 63870-000, Brazil; cyrofacundo@yahoo.com.br; 10Secretaria Municipal de Saúde de Boa Viagem (SMS-Boa Viagem), Boa Viagem, CE 63870-000, Brazil; jeanmoura2014@gmail.com (P.J.d.S.M.); fabiofelix944@gmail.com (F.F.d.S.); done4200@gmail.com (F.d.C.A.); rai_paceros@hotmail.com (R.d.S.R.); diihvale@hotmail.com (D.d.V.C.); aldessandradiniz2020@gmail.com (M.A.D.V.); 11Coordenação-Geral de Vigilância das Arboviroses (CGARB), Departamento de Imunização e Doenças Transmissíveis (DEIDT), Secretaria de Vigilância em Saúde (SVS), MS, Brasília, DF 70058-900, Brazil; aline.rapello@saude.gov.br (A.M.R.d.N.); digobh.diego@gmail.com (R.G.P.); marceloadrianoneuro@gmail.com (M.A.d.C.e.S.V.); pedro.passos@saude.gov.br (P.H.d.O.P.); daniel.ramos@saude.gov.br (D.G.R.); alessandro.romano@saude.gov.br (A.P.M.R.); 12Laboratório Central do Estado do Ceará (LACEN-CE), Fortaleza, CE 60120-002, Brazil; icavalcanteramalho@gmail.com (I.L.C.R.); ledasimoes@gmail.com (L.M.S.M.); fernandamontenegroaraujo@gmail.com (F.M.d.C.A.); 13Coordenação Geral de Laboratórios de Saúde Pública (CGLAB), MS, Brasília, DF 70723-040, Brazil; karina.cavalcante@saude.gov.br; 14Coordenação de Epidemiologia, Secretaria de Estado da Saúde do Piauí, Teresina, PI 64018-000, Brazil; 15Department of Veterinary Integrative Biosciences, Texas A&M University, College Station, TX 77843-4458, USA

**Keywords:** West Nile virus, Brazil, equids, Boa Viagem, Ceará, PRNT

## Abstract

In June 2019, a horse with neurological disorder was diagnosed with West Nile virus (WNV) in Boa Viagem, a municipality in the state of Ceará, northeast Brazil. A multi-institutional task force coordinated by the Brazilian Ministry of Health was deployed to the area for case investigation. A total of 513 biological samples from 78 humans, 157 domestic animals and 278 free-ranging wild birds, as well as 853 adult mosquitoes of 22 species were tested for WNV by highly specific serological and/or molecular tests. No active circulation of WNV was detected in vertebrates or mosquitoes by molecular methods. Previous exposure to WNV was confirmed by seroconversion in domestic birds and by the detection of specific neutralizing antibodies in 44% (11/25) of equids, 20.9% (14/67) of domestic birds, 4.7% (13/278) of free-ranging wild birds, 2.6% (2/78) of humans, and 1.5% (1/65) of small ruminants. Results indicate that not only equines but also humans and different species of domestic animals and wild birds were locally exposed to WNV. The detection of neutralizing antibodies for WNV in free-ranging individuals of abundant passerine species suggests that birds commonly found in the region may have been involved as amplifying hosts in local transmission cycles of WNV.

## 1. Introduction

West Nile virus (WNV) is one of the world’s most widespread mosquito-borne flavivirus that emerged in the Americas during late 1990s. WNV is currently a major public health and veterinarian concern worldwide as the cause of human encephalitis outbreaks, and avian and equine epizootics in temperate and tropical countries [1]. WNV is mainly maintained in nature by an enzootic transmission cycles involving passerine as amplifying hosts and *Culex* species as vectors, and human and domestic animals are dead-end hosts [2]. In Brazil, the first evidence of WNV circulation was reported in 2011 when specific neutralizing antibodies were detected in healthy horses from the Pantanal region, located in the state of Mato Grosso do Sul, west-central region of Brazil [3].

Follow-up studies indicated that WNV was probably more spread than originally thought when serological evidence of silent circulation of WNV was reported in equines from a larger area of Mato Grosso do Sul, and from the neighbor state of Mato Grosso [4,5,6,7]. The first description of WNV causing disease in the country was reported only in 2014, when WNV neutralizing antibodies were detected in a cerebrospinal fluid (CSF) sample from a patient with flaccid paralysis in the state of Piauí, northeast region of Brazil [8]. Neurologic disease associated with WNV infection in equids had never been reported until 2018, when WNV was isolated from a horse during an epizootic of neurological disease reported in the state of Espírito Santo, southeast Brazil [9,10]. In July 2019, an equine with neurological disorder was reported positive for WNV in the state of São Paulo, southeast Brazil [11]. More recently, genetic evidence of WNV in horses was also reported in Minas Gerais, a neighbor state [12].

In June 2019, a fatal case of a horse with neurological disorder was reported in a property used for equid training in the municipality of Boa Viagem, located in the state of Ceará (CE), northeast Brazil [13]. Clinical signs included locomotor disabilities, limbs paresis, pedaling movements, and mydriasis. Central nervous system samples were collected and tested negative for rabies. WNV ribonucleic acid (RNA) was detected in brain tissue by real-time reverse-transcription polymerase chain reaction (RT-PCR) at the Federal Agricultural Defense Laboratory of the Brazilian Ministry of Agriculture, Livestock, and Supply. The case was communicated to the Brazilian Ministry of Health, which led a multi-institutional task force involving several public institutions from national, state, and municipal levels, coordinated by the arbovirus surveillance team of the Brazilian Ministry of Health, for an integrated investigation of this event. The investigation included: (i) retrospective analysis of samples previously collected in CE during arbovirus and neurological syndrome surveillance programs, (ii) a survey in human, animals, and mosquito populations through eco-epidemiological and laboratory assessments, and (iii) the structuring of prospective human and animal surveillances to increase the sensitivity to detection and the response capacity. Here, we present fieldwork and part of laboratorial analysis carried out in the index case area.

## 2. Materials and Methods

In September 2019, blood samples from humans, free-ranging wild birds, equids, sheep, goats, and domestic birds, as well as adult and immature mosquitoes, were collected from several properties and households within a radius of 25 km of the probable local of infection (PLI) of the horse fatal case in the municipality of Boa Viagem (05°07′33″ S 39°43′47″ W), CE. With around 50,000 residents, Boa Viagem is located in the mesoregion of Sertōes cearenses and is part of the Caatinga biome, which has a semi-arid hot tropical climate with average temperature of 29 °C, and average rainfall of 717.7 mm concentrated from January to April. The main sources of water are rivers and water reservoirs, and the vegetation are composed of open shrubland and thorny deciduous forest, with small trees and several species of cactus [14]. Samples were collected from four main sites, including the PLI area and three local farms. For presentation purposes, the PLI area was delimited as a rectangular-shaped area of roughly two square kilometers, which includes an inner perimeter with the street where the horse index case was reported, and two surrounding neighborhoods named here as neighborhood B and C. Farms included Farm 1, which had a House sparrow communal nest site located within a radius of five km of the index case property, and Farm 2 and Farm 3 located within a radius of 25 km (Figure 1). 

The PLI area had a typical local landscape formed by open shrubland and a few artificial water collections. Local streets were unpaved and ornamented with the exotic Neem tree (*Azadirachta indica*). The property where the horse index case was reported temporarily hosts equids from other local properties to be trained for the vaquejada, which is a traditional competitive rodeo-like event in which two mounted riders pursue a steer in an open arena. All farms sampled in the present study were equestrian properties. To investigate the role of local mosquito species in the WNV transmission, immature and adult mosquitoes were collected mostly from sites within a radius of five km of the PLI. Mosquito samples were collected during two weeks in September 2019.

In November 2019, two months later, a second blood sampling from some equids and domestic birds previously sampled in September was conducted aiming to evaluate seroconversion. Blood samples were collected in tubes without anticoagulants, except for a subset of samples including most free-ranging wild birds that had blood collected in tubes with ethylenediamine tetra-acetic acid (EDTA). Blood samples were then transported in coolers with ice packs to a centralized processing station where they were centrifuged and serum and plasma removed for antibody detection. All samples were stored in liquid nitrogen in the field and, once in the laboratory stored at −70 °C for further laboratory analysis.

### 2.1. Vertebrate Sampling

#### 2.1.1. Blood Samples from Humans

Human blood samples were taken by a team of nurses and health workers from randomly selected residents of households located in the perimeter where the horse index case was reported in September 2019 (Figure 1). All participants answered a questionnaire to collect information about contact with animals, general health and living conditions, history of clinical signs, exposure to mosquitoes, and any additional observations.

All participants, including legal representatives or parents for minors, read and signed the written informed consent form. The laboratory analysis of human samples presented here were part of the Arbovirosis Surveillance Program of the Brazilian Ministry of Health, and conducted for diagnostic purposes in compliance with the requirements of Brazilian Resolution 510/2016 of CONEP (Comissão Nacional de Ética em Pesquisa), which rules on the scientific use of human samples in Brazil. The analysis conducted at Laboratório de Flavivírus da Fundação Oswaldo Cruz was approved by the Ethics Committee of Instituto Oswaldo Cruz (CAAE 90249219.6.1001.5248 #2.998.362, approval date 4 November 2018). 

#### 2.1.2. Blood Samples from Equids and Small Ruminants

Equids including horses, donkeys, and hybrids, and small ruminants including sheep and goats were sampled by a team of veterinarians in several properties located at the PLI area and in all three farms (Figure 1). All animals were submitted to a preliminary clinical evaluation with no complementary laboratory testing to assess health status. Equids were sample at the PLI area, including the perimeter with the street where the horse index case was reported and neighborhoods B and C. Equids from all three farms used to compete in vaquejadas, but only equids from Farms 2 and 3 had a history of staying at the property where the index case was reported. One of the properties located at neighborhood B had working equids used for traction and general transportation, as cart donkeys. Small ruminants were sampled in properties located in neighborhoods B and C, as well as in Farm 1. All samplings were consented by owners who also answered a questionnaire about general status of health, gender, age, tameness, breed, vaccination status, travel records, and history of abnormalities, such as clinical signs involving the central or peripheral nervous system. In November 2019, a second blood sample was collected from several equids previously sampled in September, aiming to evaluate seroconversion. Additional serum and/or brain tissue samples from the index case and two other equids that presented neurological disorder during investigations were also included for molecular analysis.

#### 2.1.3. Blood Samples from Domestic Birds

Domestic birds, including chickens, geese, and ducks, and helmeted guineafowls had blood collected also by the same team of veterinarians mostly in the perimeter where the index case was reported, but also in neighborhoods B and C. As for equids and small ruminants, domestic birds were submitted to a preliminary clinical evaluation with no complementary laboratory testing to assess health status. Some domestic birds were also blood-sampled at Farm 1 (Figure 1). When possible, domestic birds were sampled from the same properties and residences where equids and humans were sampled. All samplings were consented by owners who also answered a questionnaire about general status of health, gender, probable age, and history of abnormalities. As performed with equids, a second blood sample was collected in November 2019 from some domestic birds previously sampled in September 2019, to assess potential seroconversion.

#### 2.1.4. Blood Samples from Free-Ranging Wild Birds

Free-ranging wild birds were captured by a team of veterinarians and ornithologists using mist nets at two locations of Boa Viagem, including the PLI area and Farm 1 (Figure 1). At the PLI area, mist nets were placed roughly 150 m distant from the property where the index case was reported in a swamp area, fragments of degraded forests and open field visited and/or inhabited by various groups of birds. At Farm 1, mist nets were placed not only in a swamp area and fragments of degraded forests but also close to a gazebo with a House sparrow (*Passer domesticus*) communal nest site. Because prevalence data does not currently exist for WNV in birds of CE, for our target bird species, we aimed to sample bird species in general that are locally abundant. The use of a farm as a second sampling location was intended not only to serve as a potential control site but also to intensify the evaluation of the exposure of local common species, including House sparrow populations to WNV in Boa Viagem. Captured birds were transported within fabric bags by hand to a centralized processing station located within the collection subsite. All specimens were identified, sexed, weighted, banded, had a general status of health checked, and a blood sample taken before being released. When enough volume was available, whole blood samples were centrifuged, and plasma removed for antibody detection.

Assuming that all species are equally attractive to vector mosquitoes and that host competence is the same for all species studied, we calculated the capacity of amplification (A) as C × S^2^, where C = relative abundance and S = seroprevalence^2^. The capture rate is the prevalence of a determined species among all species captured and is a crude estimate of relative abundance of local species. C is calculated assuming that individuals of all species have similar capture probability. Seroprevalence is squared because it represents the contact rate between the infectious vector and the vertebrate host, and for amplification, vectors must contact host twice, once to infect and once to become infected. It is noteworthy that, the capacity of amplification calculated here is merely speculative. Species are often not equally attractive to vector mosquitoes, and host competence varies among different avian groups. Moreover, relative abundance estimated by prevalence in captures is likely to be biased and might not represent real relative abundance.

A subset of individuals that unfortunately died during netting or sampling had tissue samples collected and were stored for future investigations. Free-ranging birds capture and sampling were authorized by the Brazilian Federal Environmental institutions as Instituto Chico Mendes de Conservação da Biodiversidade and Sistema Nacional de Anilhamento de Aves Silvestres of the Ministry of Environment (3891/1-324582, approval date 6 May 2019), which regulates wild bird captures in Brazil.

### 2.2. Mosquito Sampling and Processing

Mosquito collections were performed by a team of veterinarians and entomologists from 4 to 10 September 2019, both in the urban and periurban area including the perimeter where the horse index case was reported and neighborhood C. Additional specimens were collected at the bank of a Boa Viagem river and Farm 1 (Figure 1). Mosquito samplings were conducted both during the daytime and at night using backpack aspirators, nets, CDC light traps, and Shannon trap mostly co-located in space and time with vertebrate blood sampling, so that data from vertebrate sampling could be linked with those from mosquito abundance and species composition. Collections were performed in several microhabitats to target diverse diurnal and nocturnal mosquito species in microhabitats where humans, domestic, and wild animals occupy and rest during the night, such as inside and in the vicinity of human dwellings, chicken sheds, and barns. Mosquitoes were also collected near water bodies such as ponds, swamps, and stream (Figure 1). All samplings conducted in private land were consented by householders or property owners. Moreover, immature mosquitoes were sampled in containers, swamps, and ponds, and reared until adult stage in the laboratory for confirmation of mosquito fauna composition based on morphological characters of immature stage and male genitalia. Mosquito collections were segregated by subsite, trap, and date, and then stored in liquid nitrogen at the field. In the laboratory, all specimens were sorted by species, sex, and blood engorgement status using a dissecting scope and dichotomous keys [15,16]. Non-engorged mosquitoes were pooled up to 12 specimens by species, sex, subsite, and date of collection. After identification, all mosquito samples were placed in cryovials and stored at −70 °C.

### 2.3. Laboratory Testing

#### 2.3.1. Serological Methods 

##### Plaque Reduction Neutralization Test (PRNT_90_)

Aliquots of serum from humans, equids, small ruminants, and domestic birds as well as aliquots of plasma from free-ranging wild birds were initially heat-inactivated at 56 °C for 30 min to inactivate proteins of the complement system and then tested by PRNT_90_ for WNV. Briefly, inactivated samples were screened by PRNT_90_ for WNV at a single 1:10 dilution, as previously described [17]. Samples that were reactive for WNV were then tested in serial two-fold dilutions that ranged from 1:10 to up to 1:20,480 for their ability to neutralize plaque formation by WNV. Exceptionally, some low volume samples were initially tested in higher dilutions. Samples that were reactive for WNV in high PRNT_90_ titers, such as ≥80, were also tested for Saint Louis encephalitis (SLEV) and Ilheus (ILHV) viruses as differential diagnosis. As additional precaution to mitigate the detection of false seropositives, human sera that were reactive for WNV were tested not only for SLEV and ILHV, but also for dengue viruses (DENV-1 and DENV-2), as previously described [7]. All infectious reference viruses used for PRNT_90_ were submitted to a commercial multiplex real-time RT-PCR kit for DENV, Zika (ZIKV), and chikungunya viruses (Bio-Manguinhos, Brazil), and also a real-time RT-PCR for yellow fever virus (YFV), as previously described [18], to confirm viral identity and discard viral contamination in viral stocks.

Serum and plasma samples were considered seropositive to WNV when it reduced at least 90% of the formation of viral plaques at a dilution of 1:20, and were seronegative for all other tested flaviviruses in monotypic reactions. Additionally, samples that were reactive for WNV and its reciprocal neutralizing antibody titer was at least four-fold greater than what was observed for the other tested flaviviruses were also considered seropositive in heterologous reactions.

Serum and plasma samples that were reactive for WNV and its reciprocal neutralizing antibody titer was not four-fold greater than what was observed for the other tested flaviviruses were considered seropositive for an undifferentiated flavivirus. Samples that presented titers of 10 for WNV and <10 for all other flaviviruses or that presented titer ≥10 for WNV but not tested for all the other flaviviruses were considered inconclusive. Samples that presented neutralizing antibody PRNT_90_ titers for WNV and all other flaviviruses <10 were considered seronegative. Finally, sera and plasmas that presented PRNT_90_ titer at least four-fold higher for any other flavivirus when compared to WNV were considered seropositive for that flavivirus. We calculated 95% confidence intervals (CIs) for seroprevalence proportions using the Wilson score method. Seroprevalence proportions were compared using the Fisher exact test.

##### Hemagglutination Inhibition Test (HI)

In parallel, aiming to evaluate the potential use of HI as a screening method for future WNV-specific antibody investigations in Brazil, aliquots of serum from humans, equids, small ruminants, and domestic birds were also tested by HI. Briefly, antigens of WNV and twelve other flaviviruses were prepared from the brain, liver, or serum of newborn mice using the sucrose-acetone extraction technique, as previously described [19,20]. Human samples were tested for 11 flaviviruses, and domestic animal samples were tested for seven flaviviruses (Appendix A). 

Sera were considered seropositive for WNV in monotypic reaction, when at a 1:20 dilution reacted only against WNV antigen. Samples were also considered WNV-seropositive in heterotypic reactions, when the HI titer for WNV is at least four-fold higher when compared to HI titer of all other flaviviruses tested, as previously described [21]. Serum and plasma samples that were reactive for WNV but the HI titer was less than four-fold greater for WNV than to the other flaviviruses were considered seropositive for an undifferentiated flavivirus. Samples that presented HI titer ≥20 for WNV but not fully tested for the other flaviviruses were considered inconclusive. Sera that presented HI titer at least four-fold higher for any other flavivirus when compared to WNV were considered seropositive for that flavivirus. Finally, samples that presented HI titer <20 for WNV and all other flaviviruses were considered seronegative.

Aiming to identify an alternative criterion of selection using HI as a screening method for PRNT_90_, in a secondary analysis, named screening HI B, we considered seropositive by HI all samples that presented titer ≥20 for WNV regardless the titers for other flavivirus antigens. Additionally, as a complementary survey to assess local exposure of hosts to other arboviruses, serum samples from humans and most domestic animals were also tested for other 14 nonflavivirus antigens, including alphaviruses and orthobunyaviruses (Appendix A).

##### IgM Antibody Capture Enzyme-Linked Immunosorbent Assay (MAC-ELISA)

Serum samples from humans from Boa Viagem were also tested by MAC-ELISA for WNV and SLEV. MAC-ELISA detects viral specific immunoglobulin M (IgM) at early stages of a primary infection. In brief, microplate wells were coated with IgG anti-human IgM, and then sample, antigen, and substrate were successively added to the system. Reaction was chemically stopped and plates read in a spectrophotometer. A positive test result was obtained when the P/N of the test serum was 2.0, and the mean of the *A*_450_ values of the test serum reacted on viral antigen was at least twice the mean of the *A*_450_ values of serum reacted on normal mouse brain antigen. When the latter criterion was not met due to nonspecific reaction with the normal mouse brain antigen, the result was reported as uninterpretable, as previously described [22,23,24].

#### 2.3.2. Molecular Methods

##### Real-Time Reverse-Transcription Polymerase Chain Reaction (RT-PCR)

A subset of serum samples of humans and equids were submitted to RNA extraction followed by real-time RT-PCR for WNV and SLEV based on the amplification of a region of the envelope gene, as previously described [25,26]. Additionally, brain tissues from the index case and two more equids that presented neurological disorder during investigations were also tested by real-time RT-PCR. All samples were processed and inactivated in a biosafety level 3 laboratory (BSL3) and then tested by real-time RT-PCR in BSL2 facilities. All reactions were ran with positive and negative controls. Nucleic acid extraction was performed in a KingFisherFlex Automatic Extractor (Termo Fisher Scientifc, Waltham, DC, USA) using a MagMAXTM Pathogen RNA/DNA kit (Life Technologies, Carlsbad, CA, USA) in accordance with the manufacturer’s instructions. Real-time RT-PCR included also mosquitoes collected in Boa Viagem in September 2019. Mosquito pools were homogenized in 250–500 µL of L−15 culture medium by using the Precellys 24^®^ tissue homogenizer with sterile glass grinding beads. After clarification by centrifugation (9600× *g*, 10 min, 4 °C), a 140 µL aliquot of the supernatant was removed for RNA extraction using QIAamp Viral RNA Mini Kit (Qiagen, Hilden, Germany) according to the manufacturer’s instructions. RNA samples were then tested for WNV by a specific real-time RT-PCR, as previously described [26].

##### Nucleotide Sequencing by Illumina

Samples that tested positive for WNV by RT-PCR were submitted to nucleotide sequencing. In brief, positive samples had RNA reextracted and retested by RT-PCR with specific primers designed to amplify a ~500 nucleotide fragment (unpublished data). PCR product was purified and sequenced by using Illumina sequencing chemistry using Nextera DNA Flex Library Prep (Illumina, San Diego, CA, USA) according to the manufacturer specification. The size distribution of the libraries was evaluated using a 2100 Bioanalyzer (Agilent, Santa Clara, CA, USA), and the samples were pair-end sequenced (2 × 300 bp) on a MiSeq v3600 cycle (Illumina, San Diego, CA, USA). The FASTQ reads obtained were assembled, trimmed, and mapped by Geneious 11.0.3 (Biomatters, https://www.geneious.com) (accessed on June 2021) against a reference sequence of WNV (MT_90560) from the state of Espírito Santo, Brazil [10] available in Pubmed database (https://www.ncbi.nlm.nih.gov/nuccore/?term) (accessed on June 2021).

## 3. Results

In September 2019, fieldwork teams assessed 80 human residents, captured 379 free-ranging wild birds, sampled 25 equids, 34 sheep, 31 goats, 67 domestic birds, and collected 853 adult mosquitoes in multiple sites within a radius of 25 km of the PLI in Boa Viagem.

A total of 513 individuals, including 78 humans, 278 free-ranging wild birds, and all equids, small ruminants, and domestic birds sampled were tested by PRNT_90_ for WNV. For comparative analysis, 233 of these samples were also tested by HI, and all human samples tested by MAC-ELISA as well. In November 2019, a second blood sampling from 18 equids and 39 domestic birds was also taken and tested by PRNT_90_ aiming to evaluate seroconversion.

Vertebrate samples tested by real-time RT-PCR included sera of all 25 equids and 78 humans sampled in September 2019, as well as a subset of extra samples, including equid tissue samples from the index case, and two other equids that presented neurological disorder during the investigation.

Regarding mosquitoes, a total of 853 adult individuals of 22 species belonging to seven genera and 11 subgenera were captured and pooled from 15 subsites (Figure 1), and all of them tested for WNV by real-time RT-PCR.

### 3.1. Human Samples

In September 2019, human serum samples were taken from 78 residents of 40 households located at the PLI area (Figure 1). From 78 serum samples tested, 14 (18%) were selected at the screening PRNT_90_ for presenting titer ≥10 and two (2.6%) from residents within the PLI area presented a monotypic reaction to WNV by PRNT_90_ endpoint titer (Figure 2). Seropositive samples were from a ten year old boy and an eight year old girl from households located at the same street and near to the property where the index case was reported (Table 1). Residents surveyed were asked through a questionnaire about epidemiological and clinical conditions from April 2019, when the first cases of horses with neurological disorders were reported. Both WNV-seropositive individuals referred to be bitten by mosquitoes at night. The eight year old girl reported contact with a dead horse in July 2019 and consumption of free-ranging Eared dove (*Zenaida auriculata*) meat during this period. The boy reported illness during the period, including fever of 39 °C and sore throat. He was diagnosed with tonsillitis, and fully recovered with no need for hospitalization. The two individuals that were seropositive for WNV by PRNT_90_ were nonreactive for WNV or SLEV by MAC-ELISA, and among the four samples that were reactive for WNV by MAC-ELISA, three were seronegative and one inconclusive for presenting a titer of 10 for WNV by PRNT_90_ (Appendix A). Sera from all 78 humans sampled in September 2019 in Boa Viagem, including the two WNV-seropositive children tested negative for WNV and SLEV by real-time RT-PCR.

### 3.2. Equid Samples

A total of 22 horses, two donkeys, and one male hybrid were collected from six properties of Boa Viagem (Figure 1). All animals appeared healthy at the moment of venipuncture. In November 2019, a second blood sample was collected from 18 equids of six properties previously visited in September, aiming to evaluate potential seroconversion. From all 25 equid samples collected in September 2019, 20 were selected at the screening PRNT_90_ for presenting titer ≥10, and 11 (44%) confirmed seropositive for WNV (Figure 2), with seven presenting monotypic reactions, and four heterotypic reactions by PRNT_90_ endpoint titer. Samples with heterotypic reactions presented PRNT_90_ titer four-fold higher for WNV than for SLEV and ILHV. WNV-seropositive equids were sampled at the index case street (*n* = 2), neighborhood B (*n* = 3), Farm 1 (*n* = 3), Farm 2 (*n* = 2), and Farm 3 (*n* = 1) (Table 2). Among the equids that had a second serum sample collected in November 2019, none presented a four-fold increase in PRNT_90_ titers when compared to the results from samples collected in September, but three equids went from PRNT_90_ titer <10 to ≥10 (Table 3). In total, 5 out of 11 (45%) WNV-seropositive equids had a history of recent stay at the case index property for training.

Regarding molecular testing, except for the brain tissue of the index case that was confirmed positive for WNV by real-time RT-PCR, all 25 serum samples and the additional subset of tissue samples from the two other equids that presented neurological disorder during investigations tested negative by real-time RT-PCR for WNV and SLEV. From the index case, a small fragment of 497 nucleotides (NCBI Reference Sequence: MZ_557450) of WNV genome was recovered and sequenced by Illumina Miseq. High identity scores were obtained with WNV lineage 1a sequences. The WNV sequence from Boa Viagem (MZ_557450) presented 99.8% similarity with a Brazilian isolate (MT_90560) from the state of Espírito Santo [10], and 99.4% with the ancestral isolate NY99 (AF_196835.1) [13].

### 3.3. Domestic Birds

A total of 67 domestic birds, including 63 chickens, two goose, one duck, and one helmeted guineafowl from 14 properties or households were sampled not only at the PLI area but also at Farm 1 (Figure 1). All animals were apparently healthy at the moment of venipuncture, and from sera of 67 individuals, 22 (33%) were selected at the screening PRNT_90_ for presenting titer ≥10. Of 22 domestic birds that were reactive in the screening for WNV, four roosters, eight chickens, a goose, and a nonrecorded species (20.9%) were confirmed seropositive for WNV (Figure 2). Seropositive birds were from eight (57%) out of 14 households located at the index case street of the PLI area, neighborhood C, and Farm 1 where domestic birds were sampled (Table 4).

In November 2019, a second blood sample was collected from 39 domestic birds of 12 properties, which were previously sampled in September to assess potential seroconversion. Stipulated as the four-fold increase of PRNT_90_ titer between the first and second sample, seroconversion was observed in four domestic birds (Table 5), including one rooster, two chickens, and one female helmeted guineafowl from three households located at the perimeter where the horse index was reported (Figure 1). 

Additionally, three serum samples that were collected in November 2019 presented high PRNT_90_ titers for WNV of 1280, 2560, and 5120. Unfortunately, the animal identification of these samples could not be confirmed during the second blood collection. However, except for one individual that had PRNT_90_ ≥320, the other two WNV-seropositive birds had titers equal or smaller than 320 for WNV in the first sample. These findings indicate that at least two more individuals seroconverted with a four-fold increase in PRNT_90_ titers between September and November of 2019.

### 3.4. Free-Ranging Wild Birds

Bird trapping effort was 39,420 m^2^.h of mist netting in each one of the two sampling locations in Boa Viagem (Figure 1). A total of 379 free-ranging wild birds of 42 species were captured (Appendix A). Of these, 264 specimens of 38 species were captured in the surroundings of PLI, and 115 specimens of 18 species were captured at Farm 1 (Appendix A). No clinical features suggestive of potential WNV infection were observed in the sampled birds, and the most captured specimens were House sparrow (*n* = 85, 22.4%), followed by Picui ground dove (*n* = 50, 13.2%) and Blue-black grassquit (*n* = 36, 9.5%) (Appendix A). Among bird specimens captured at only the PLI area, most common species were Picui ground dove (*n* = 43, 16.3%), Blue-black grassquit (*n* = 35, 13.2%), and Sayaca tanager (*n* = 20, 7.6%). Regarding the bird specimens captured only at Farm 1, most prevalent were House sparrow (*n* = 68, 59.1%), Grey pileated finch (*n* = 9, 7.8%), and Picui ground dove (*n* = 7, 6.1%) (Appendix A).

From 379 birds captured, 278 individuals of 38 species were tested by PRNT_90_. From 274 samples screened for WNV by PRNT_90_, 60 (21.9%) were selected for presenting titer ≥10. Four samples that were tested by endpoint PRNT_90_ were not previously submitted to the screening test. A total of 13 (4.7%) individuals of eight species were seropositive for WNV for presenting heterotypical reaction, with PRNT_90_ titers four-fold higher for WNV than for SLEV and ILHV (Table 6). WNV-seropositive birds were found in both sites, including 5 out of 176 (2.8%) from the PLI area, and 8 out of 102 (7.8%) from Farm 1 (*p*-value 0.076), and the PRNT_90_ titers for WNV ranged from 80 to 1280 (Table 6).

Among the species with more than five specimens tested, the highest seroprevalences for WNV by PRNT_90_ were observed in Rufous-bellied thrush (33.3%), Ruddy ground dove (15.4%), Band-tailed hornero (14.3%), Sayaca tanager (10.5%), Grey pileated finch (9.1%), and House sparrow (3.9%) (Table 7). The following species Picui ground dove, Blue-black grassquit, Plain-breasted ground dove, Great kiskadee, Black-backed water tyrant, Smooth-billed ani, Yellow-chinned spinetail, and White-throated seedeater had more than five individuals tested, and all were seronegative for WNV.

Among all species captured in both sites, capacity of amplification would be higher for Orange-headed tanager followed by Rufous-bellied thrush, Ruddy ground dove, Blue-winged parrotlet, Sayaca tanager, Band-tailed hornero, House sparrow, and Grey pileated finch (Table 8). When considering only the four species with more than 10 specimens tested, the capacity of amplification would be higher for Ruddy ground dove followed by Sayaca tanager, House sparrow, and Grey pileated finch (Table 8). If considering only the birds captured at the PLI area, capacity of amplification would be higher for Blue-winged parrotlet followed by Rufous-bellied thrush, Sayaca tanager, and Ruddy ground dove (Appendix A). Regarding specimens captured only at Farm 1, the capacity of amplification was higher for Orange-headed tanager followed by Ruddy ground dove, Rufous-bellied thrush, Band-tailed hornero, House sparrow, and Grey pileated finch (Appendix A). 

### 3.5. Small Ruminants

Blood samples from 65 small ruminants, including 34 sheep and 31 goats were taken from four local properties at the PLI area and Farm 1 (Figure 1). All animals were apparently healthy at the moment of venipuncture, and from a total of 34 sheep and 31 goats of four different properties collected in September 2019, two goats and one sheep (4.6%) were selected at the screening PRNT_90_ for presenting titer ≥10. Of these, previous exposure to WNV was confirmed in one goat (1.5%) by the detection of specific neutralizing antibodies in heterotypic reaction (Figure 2). The WNV-seropositive goat presented PRNT_90_ titer of 80 for WNV, 20 for SLEV and <10 for ILHV (Appendix A). The seropositive goat was sampled at a property located at neighborhood B, distant roughly 1.5 km from the property where the index case was reported.

### 3.6. Mosquitoes 

A total of 853 adult individuals of 22 species belonging to seven genera and 11 subgenera were captured and tested negative for WNV by real-time RT-PCR. Mosquitoes of genus *Culex* (45%) and tribe Mansoniini (genera *Mansonia* and *Coquilettidia*) (48.5%) accounted for 93.6% of this total (Table 9). Nearly 48% of captured females were Mansoniini, where *Mansonia titillans* was the most frequent with 28.1% of total caught females, followed by the domestic species *Culex quinquefasciatus* (13.7%). 

Altogether, *Culex* species accounted for 41.8% of captured females. Strictly ornithophilic mosquitoes such as *Aedeomyia squamipennis* and the identified zoophilic anophelines were, respectively, only 0.9% and 5.5% of the total adults captured. Considering the environment and collection method, 85.1% of anopheline mosquitoes were caught at the farm, contrasting with the urban restricted distribution of *Aedes aegypti* (Appendix A). 

The mosquito fauna composition was much richer in the periurban area, which includes PLI and where 96.1% of *Cx. quinquefasciatus* were captured. This species accounted for 21% of the total mosquitoes captured in the PLI and surroundings in the periurban environment, surpassed only by *Ma*. *titillans,* which together with the other Mansoniini species totaled 43.7%. Nearly 79% of *Cx. quinquefasciatus* were captured inside houses located either in the farm or in the urban area. Except for this species, the other *Culex* species were by far more frequent on the open fields (88.3%) and inside animal shelters (9.9%) close to houses (Appendix A). 

By splitting the total number of mosquitoes caught according to the location and method of capture, it was found that most adults (61.1%) were found on the open fields, where 43.3% were captured with human attraction and the rest with CDC light traps. Only 18.4% of mosquitoes were collected in animal shelters near houses, 64.9% and 35% of which were found inside chicken sheds and horse barns, respectively. Around 73% of captured mosquitoes in the horse barns belonged to at least five out of the nine species of genus *Culex* detected in the PLI and surroundings (Appendix A).

The ponds neighboring the PLI area were the larval sites of at least six Culex species (*Cx. quinquefasciatus*, *Cx. chidesteri*, *Cx. coronator* Complex, *Cx. declarator*, *Cx. inhibitator*, *Culex* (*Mel.*) sp. and *Cx. panacossa*), *Ma. titillans* and *Anopheles triannulatus* s.l. (data not shown). *Cx. quinquefasciatus* was growing in containers inside houses and in drinking fountains for horses, a larval habitat shared with *Cx. declarator*, which was the only species found in flood areas at the edge of the river in Boa Viagem (Figure 1). 

### 3.7. Evaluation of HI as Screening Method for PRNT_90_ for WNV

#### 3.7.1. Human Samples

From all 78 human sera tested by both PRNT_90_ and HI, two (2.6%) were confirmed seropositive for WNV by PRNT_90_ and one (1.3%) by HI (Figure 3). The two samples that tested seropositive by PRNT_90_ presented monotypic reactions for WNV. When tested by HI, one was also seropositive for WNV for presenting four-fold greater titer for WNV when compared to the other flavivirus antigens tested, and the other one was seronegative for WNV and considered seropositive for an undifferentiated flavivirus for presenting reactivity to other flavivirus antigens (Appendix A). From 78 samples tested by both methods, 64 samples were seronegative for WNV by PRNT_90_, and of these, 11 (17.2%) were also seronegative for WNV by HI. The remaining 53 samples (83%) were considered seropositive for an undifferentiated flavivirus for presenting HI titer ≥20 not only for WNV, but also for several other flaviviruses, and with less than four-fold titer difference among them (Appendix A).

If considering screening HI B, which includes all samples that presented HI titer for WNV ≥20 as a potential alternative screening criterion, 66 (85%) samples would be selected for endpoint PRNT_90_ for WNV, but as observed using the conventional seropositivity criterion, only one (50%) out of the two samples would be confirmed seropositive for WNV by endpoint PRNT_90_ (Figure 3).

#### 3.7.2. Equid Samples

From 25 equid samples tested for WNV by PRNT_90_ and HI, 11 (44%) were seropositive for WNV by PRNT_90_, and four (36.3%) of these were also seropositive for WNV by HI (Figure 3). The remaining seven samples were seropositive for an undifferentiated flavivirus (*n* = 6) or seronegative (*n* = 1) by HI (Appendix A). All the samples that were seropositive for WNV by HI presented titer of 20 in monotypic reactions. Of a total of five samples that were seronegative by PRNT_90_ for WNV, four (75%) were also seronegative by HI. The remaining sample was seropositive for an undifferentiated flavivirus (Appendix A). When considering all samples that presented HI titer for WNV ≥ 20 as screening criterion, 19 (76%) samples would be selected for PRNT_90_ for WNV, and of these, 10 (91%) out of 11 samples, would be later confirmed seropositive for WNV by PRNT_90_ (Figure 3).

#### 3.7.3. Domestic Birds

Regarding serum samples from 65 domestic birds that were tested by both serological methods, 14 (21.5%) were confirmed seropositive for WNV by PRNT_90_. Of these, four (28.6%) were also seropositive for WNV by HI (Figure 3). The remaining ten samples were seropositive for an undifferentiated flavivirus (*n* = 8) or seronegative (*n* = 2). From 47 samples that tested seronegative for WNV by PRNT_90_, 46 (98%) were also seronegative by HI, and one was seropositive for WNV (Appendix A). When considering all samples that presented HI titer for WNV ≥ 20 as screening criterion, 16 (24.6%) samples would be selected for endpoint PRNT_90_ for WNV. Using this criterion, 12 (75%) would be later confirmed seropositive for WNV by endpoint PRNT_90_ (Appendix A).

#### 3.7.4. Small Ruminants

From 65 small ruminants tested by both methods, one goat (1.5%) was seropositive for WNV by PRNT_90_, with heterotypic reaction (Figure 3). When tested by HI, only two female goats, including the one seropositive by PRNT_90_, were reactive but both were considered inconclusive, as presented equal HI titers for WNV, SLEV, and ILHV (Appendix A). From 64 samples that tested seronegative for WNV by PRNT_90_, 63 (98%) were also seronegative by HI. If considering all samples that presented HI titer for WNV ≥ 20 as screening criterion, the same two (3.1%) samples would be selected for PRNT_90_ for WNV, and of these, one (50%) out of two samples, would be later confirmed seropositive for WNV by PRNT_90_ (Figure 3).

## 4. Discussion

### 4.1. Circulation of WNV in Boa Viagem

Epidemiological data associated with clinical, molecular, and serological diagnosis are the most reliable approach to confirm arbovirus infection. Findings presented here, including the molecular confirmation of the clinical case and the detection of highly specific neutralizing antibodies in various local vertebrate hosts, including potential amplifying bird species, confirm the past circulation of WNV in the municipality of Boa Viagem, CE.

Except for ovine, all other vertebrate groups locally investigated, including humans, equids, domestic, and wild birds, and goats had at least one individual that tested seropositive for WNV by a highly specific serological test. Higher prevalence for WNV was observed in equids and birds when compared to humans and small ruminants, and the seroprevalence varied greatly among species, going from 44% in equids to 1.5% in small ruminants (Figure 2). The variation of WNV exposure by host groups has been reported elsewhere. In a study conducted in the United States (USA) after the 1999 epidemic in New York City, seroprevalence in birds reached 50% when compared to 2.6% in humans [27].

The infection by WNV in the brain tissue of the index case was confirmed by real-time RT-PCR, and 5 out of 11 (45%) WNV-seropositive equids had a confirmed history of staying at the PLI area at some point. These findings corroborate the role of the PLI area as a source of infection in Boa Viagem. Interestingly, the area where the index case was reported has a pond that used to be surrounded by trees and accessed by the equids after training. The area was recently cleared for settlement, and the niche impact for the surge of WNV horse clinical infection in Boa Viagem needs further investigation.

### 4.2. Exposure to WNV in Boa Viagem

#### 4.2.1. Humans

In the present study, WNV-seropositive samples were collected three months after the index case was reported, mostly from properties located within a five km radius of the index property. The two seropositive humans of Boa Viagem whose were sampled in September 2019 lived in the same street where the index property was located, and none of them tested positive for WNV either by real-time RT-PCR or MAC-ELISA or described specific clinical signs suggestive of acute WNV infection during sample collection. On the other hand, the analysis of their answers to the questionnaire suggests that both of them may have been potentially exposed to WNV from April to September 2019. Both of them described nocturnal mosquito bites during this period, and the eight year old girl described contact with a dead horse in July 2019. Moreover, the ten year old boy presented fever and sore throat and was diagnosed with tonsillitis during this period. These results suggest the potential exposure of humans to WNV in Boa Viagem before the horse case was reported. Evidence of silent circulation of mosquito-borne arboviruses prior clinical cases has been reported elsewhere [28]. 

The evidence of WNV exposure in residents of Boa Viagem is probably underestimated. From 78 residents tested by PRNT_90_, two (2.6%) presented a monotypic reaction to WNV. Although, two other samples that were seropositive for SLEV and DENV-1 had PRNT_90_ titer of 80 for WNV, and ten (12.8%) others presented PRNT_90_ titers of 10. Because of the low titer, these last samples were not further tested, and for that reason, their seropositivity for WNV was never confirmed. These findings combined indicate that other residents of Boa Viagem could have been potentially exposed to WNV. Besides, restraining case confirmation by the detection of specific neutralizing antibodies using PRNT_90_ is instrumental for highly specific results, but the exposure to WNV that results in non-neutralizing antibodies may not be detected. From all 78 residents tested by MAC-ELISA, 12 (15.4%) were reactive or borderline for WNV, and all of them were seronegative or inconclusive by PRNT_90_ (Appendix A). 

On the other hand, the use of laboratory tests not designed to detect specific neutralizing antibodies in areas known for the circulation of several flaviviruses may complicate the interpretation of serological results [29]. In the present study, serological evidence of at least two other flaviviruses was observed in residents of Boa Viagem. During differential diagnosis, one person was found seropositive for SLEV and another one for DENV-1 (Appendix A). Although this evidence cannot assure these residents were locally exposed, neutralizing antibodies for SLEV were also detected in equids and free-ranging birds. Therefore, at least two other flaviviruses may have potentially circulated in Boa Viagem. Considering that the circulation of other flaviviruses was already expected in the region, SLEV and ILHV were included as differential diagnosis for all samples, and DENV-1 and DENV-2 for human samples.

Additional flaviviruses as ZIKV, could also have been included as differential diagnosis. However, according to the State Health Department of CE, from 78 suspected cases of Zika fever reported in CE from January to August of 2019, only five (6.4%) were confirmed by laboratory testing, and none of them were from Boa Viagem [30]. Because of the limited volume of serum samples and the low circulation of ZIKV in Boa Viagem during the sampling, DENV-1 and DENV-2, which were the most prevalent flaviviruses in CE in 2019, were preferred as differential diagnosis for human samples.

The low circulation of ZIKV in the region was corroborated by HI results of human samples. Among the antigens of 11 flaviviruses, ZIKV was the antigen that had the highest number (*n* = 28, 36%) of nonreactive human samples. Moreover, the two samples that were confirmed seropositive for WNV by PRNT_90_, one of them was nonreactive for ZIKV, and the other one was seropositive for WNV by HI (Appendix A). 

Another potential source of heterologous antibodies could have been the exposure to the YFV. Serum samples from the two individuals that were seropositive for WNV by PRNT_90_ were also tested by HI for both sylvatic and vaccinal antigens of YFV. One of them was seropositive for WNV, and the other one was seropositive for an undifferentiated flavivirus by HI (Appendix A). 

#### 4.2.2. Equids and Domestic Birds

Despite 44% of equids sampled in September 2019 were seropositive for WNV, among the 20 equids that had a second serum sample collected in November 2019, none presented seroconversion by the four-fold increase in PRNT_90_ titers (Table 3). The absence of seroconversion might be explained partially by the high prevalence among all equids tested and also the high PRNT_90_ titers observed in samples collected in September 2019. About 80% of the equids sampled in September 2019 presented neutralizing antibodies for WNV, and the average of their PRNT_90_ titers was 216.4. The same animals sampled in November presented an average of PRNT_90_ titers of 174.3. The lower titer average observed in November could indicate the natural decrease of antibody titers along time, or the reduction and even interruption of WNV circulation after the fatal case. It is noteworthy that three equids went from PRNT_90_ titer for WNV < 10 to 10 and 20 (Table 3), but because of the limited variation, seroconversion could not be confirmed.

The index case was reported in June of 2019, which is the beginning of winter and dry season in Boa Viagem. Therefore, a reduction in mosquito population density by September 2019 was expected. Interestingly, the precipitation in Boa Viagem was between 600–800 mm in 2019, and March of that year had the third highest precipitation in the last 20 years [31]. The high precipitation followed by the dry season could be related to the spillover event to the affected horse followed by the absence for another clinical case. During the investigations between September and November of 2019, two other neurological disorder cases in equids suspected of rabies were reported in Boa Viagem, but both of them tested negative for WNV by real-time RT-PCR.

In the present study, domestic birds presented the second highest seroprevalence for WNV (20.9%) among all vertebrate groups tested (Figure 2). Confirmed WNV-seropositive domestic birds were from eight (57%) households located at the index case street of the PLI area, neighborhood C, and Farm 1 (Figure 1). A total of four roosters, eight chickens, a goose, and a nonrecorded species (20.9%) were confirmed seropositive for WNV, and at least 12 (86%) presented monotypic reactions (Table 4). Although showing no clinical signs, and viremia unlikely to infect *Culex* mosquito vectors, chickens when infected by WNV can mount specific humoral response, and for that reason they have been used as useful sentinels for WNV surveillance programs worldwide [32,33].

#### 4.2.3. Free-Ranging Wild Species of Birds

Passerine birds tend to be highly competent for infecting mosquitoes with WNV [34], and we report here evidence of WNV exposure in different passerine species collected in both sampling sites of Boa Viagem, CE. A wide variety of passerine species was captured during the investigation, but for most species, a reduced number of specimens was sampled (Appendix A). Among the species of birds captured at the PLI area, roughly 75% had fewer than ten specimens collected. In Farm 1, around 90% of the species captured had fewer than ten specimens collected. Specimens from a total of eight free-ranging bird species were seropositive for WNV in Boa Viagem (Table 6). WNV-seropositive animals were detected in both sites, but the difference in seroprevalence was not statistically significant. Among them, Rufous-bellied thrush, Ruddy ground dove, Band-tailed hornero, Sayaca tanager, Grey pileated finch, and House sparrow presented the highest seroprevalence among species with at least six individuals tested (Table 7). Among other species with at least one WNV-seropositive individual are Blue-winged parrotlet with two individuals tested and one seropositive, and Orange-headed tanager with the only individual tested being seropositive (Appendix A). Since mortality has not been reported for any species in Boa Viagem, seroprevalence can be related to infection rate. A biased low prevalence could be found in a highly susceptible species that infected individuals fatally succumb to infection and consequentially are not captured. However, despite the detection of seropositivity among various species of wild birds, it is noteworthy that seropositivity rates alone do not reflect a bird population’s force of infection.

Relative abundance of a bird species in concert with seroprevalence is needed to identify avian amplifying hosts for WNV. It is noteworthy that amplification is speculative and only with quantitative experimental data on viremia to better understand which species are important amplifying hosts [27]. 

Unfortunately, the relative abundance of the bird species captured in Boa Viagem is unknown, and no data about WNV viremia in these species is available. Despite uncommon birds may be infected and may transmit viruses, only abundant birds are important amplifying hosts [35]. By the current knowledge about enzootic cycles of transmission of WNV worldwide, to play a role as local amplifying host the ideal species would be abundant and a ubiquitous passerine.

#### 4.2.4. Small Ruminants

In the municipality of Boa Viagem, caprine and ovine herds are economically important and commonly seen roaming through unpaved streets of the periurban areas. In the present study, all sheep were seronegative, and only one goat presented a heterotypic reaction for WNV. These findings suggest that small ruminants from Boa Viagem were either less exposed to WNV vectors, or these hosts did not mount an effective humoral immune response for WNV. Studies have shown that sheep seem to be less likely to become infected through mosquito transmission in some regions of Brazil due to their low attractiveness to mosquito bites compared to human and other domestic animals. Comparative evaluations carried out in a locality in the southeast Brazil with similar mosquito fauna showed that sheep attracted fewer mosquitoes than rooster, human, cow, and horse. Sheep were also bitten by fewer species of *Culex* when compared to the other animals [36]. In a large serosurvey conducted in a WNV transmission area in west-central Brazil, over 230 sheep tested negative for flaviviruses by blocking-ELISA, and two of them presented low PRNT_90_ titers for WNV of 20 and 10 [7]. 

#### 4.2.5. Mosquito Populations

It is noteworthy that among all collected mosquitoes, *Culex* species, which are the main WNV vectors worldwide, accounted for, respectively, 45% and 38.4% of total and female mosquito collected in Boa Viagem (Table 9). Moreover, *Culex* species were 73% of mosquito caught in horse barns (Appendix A). While only by direct diagnostic methods would be possible to confirm active circulation of WNV in Boa Viagem, the occurrence of ornitophilic and opportunistic mosquito species describes the receptivity for WNV-vectored transmission in Boa Viagem. In fact, the various ponds, flooded margins of a river in Boa Viagem, and the horse waterer that is always kept filled have seemed to be perennial *Culex* larval habitats even during the drought period. The swamp located about 150 m from the property where the index case was reported showed to be a productive adult source of *Culex* and Mansoniini mosquitoes in September 2019 (Appendix A). 

Although determining the vectorial capacity for WNV requires studies of vector competence, and field estimates of virus infection, relative abundance, and host selection [37], among 22 mosquito species captured in Boa Viagem, five (22%) of them (Appendix A) have been previously found positive for WNV in the USA [38]. 

*Culex (Cux.) quinquefasciatus*, which has been found to be moderately vector-competent for WNV in laboratory experiments [39,40], was the second most prevalent mosquito species in Boa Viagem. Despite being abundant and a WNV vector elsewhere, vector competence studies with the Brazilian populations of not only *Culex (Cux.) quinquefasciatus* but also other species of *Culex* are needed to confirm involvement of this species in WNV transmission in the country. In Guatemala, WNV was isolated from *Culex quinquefasciatus* and to a lesser extent, from *Cx. mollis*/Cx. *inflictus*, but not from the most abundant *Culex* species, *Culex nigripalpus* [41]. 

Additional testing, as the determination of the proportion of mosquito bloodmeals from the most prevalent *Culex* species taken from different bird species, could contribute to a better understanding of the local vector–host relationships in Boa Viagem. The identification of host DNA by the detection of the mitochondrial COI gene and/or cytochrome B genes [42] in blood-engorged mosquitoes has been associated with vector infection rates and serological data to identify enzootic cycles of transmission of WNV in the USA [43,44].

### 4.3. Potential Amplifying Hosts of WNV in Boa Viagem

Although the main amplifying bird species of WNV in Boa Viagem remains to be confirmed, multiplying the squared seroprevalence per species by their capture prevalence, we grossly speculated amplification capacity of the captured species in Boa Viagem. Because of the bias of bird sampling in Farm 1, which had as one of the targets a House sparrow communal nest site, we considered three scenarios, including both sites together (Table 8), PLI alone (Appendix A), and Farm 1 alone (Appendix A). Combining results, stronger amplification capacity was observed for Orange-headed tanager and Rufous-bellied thrush and Ruddy ground dove. When considering only species with more than ten specimens captured, Ruddy ground dove and Sayaca tanager presented stronger amplification capacity.

Similar results were seen when analyzing only captures from the PLI area. While stronger amplification capacity was observed for Blue-winged parrotlet and Rufous-bellied thrush, among the most abundant species, Sayaca tanager and Ruddy ground dove presented the highest amplification capacity (Appendix A). 

At Farm 1, Orange-headed tanager and Ruddy ground dove had the higher amplification capacity, and the House sparrow that was the most captured species presented 4.8% seroprevalence for WNV (Appendix A). Any of these aforementioned species are candidates for WNV amplification in Boa Viagem. It is possible that more than one of these species played a role in WNV amplification in Boa Viagem. In Louisiana, which has a humid subtropical climate, a variety of passerine bird species combined play an important role as amplifying hosts in the WNV transmission cycles [45].

Of the potential amplifying hosts of WNV in Boa Viagem, only the Ruddy ground dove is nonpasserine, which is seen to be likely less competent to infect blood-feeding mosquitoes elsewhere. However, other studies have demonstrated that doves are being exposed to WNV and may contribute to the mass of WNV-infectious mosquitoes in some regions as Colorado in the USA [42]. In Argentina, free-ranging specimens of White-tipped Dove (*Leptotila verreauxi*) were found seropositive for WNV [46], and in experimental infections, Picui ground doves presented the highest and longest viremia among three autochthonous bird species evaluated. These results suggest that Picui ground dove is a potential amplifying host for WNV in Argentina [47]. 

Serological evidence of WNV infection has also been reported in Ruddy ground dove from Guatemala and Mexico [41,48]. In a study conducted in different regions of Brazil between 2008 and 2010, 24 Ruddy ground doves were tested, but no evidence of WNV infection was reported [6]. In the present study, 2 (15.4%) out 13 individuals tested were seropositive for WNV and Ruddy ground dove presented the third highest amplification capacity in Boa Viagem (Table 8). Considering the abovementioned evidence and its wide geographical distribution, Ruddy ground doves and other species of columbiformes should not be ruled out as potential amplifying hosts of WNV in Boa Viagem or elsewhere in the country.

Although Rufous-bellied thrush and the Band-tailed hornero presented high seroprevalence for WNV among all species with a sample size of *n* > 5 (Table 7), these species were not abundant, corresponding for less than 2% each of all birds captured. Evidence of these species acting as amplifying hosts of WNV worldwide remains unreported, and most evidence is restricted to eventual serological findings of WNV exposure in closely related species. Rufous hornero, which is a species of Furnariidae, as the Band-tailed hornero, has been exposed to WNV, and if proven to be competent amplifying host, could play role in WNV transmission in Argentina [49]. Regarding the Rufous-bellied thrush, of four individuals sampled in Buenos Aires between 2012 and 2013, none of them were seropositive for WNV [50]. In Brazil, an investigation for WNV conducted between 2008 and 2010, 18 Rufous-bellied thrush tested negative for WNV by molecular and serological methods [6]. Between 2012 and 2013, free-ranging birds, including 13 individuals of Rufous-bellied thrush captured from green areas of the city of São Paulo, southeastern Brazil, tested negative for flaviviruses by real-time RT-PCR [51].

Other WNV antibody-positive trushes and congeners have been eventually detected in Central and South America [52,53,54]. Serological evidence of WNV infection was detected in pale-breasted thrush in Venezuela [55]. In Guatemala, Clay-colored thrushes were abundant in a WNV transmission area but were not considered competent amplifying hosts of WNV [41]. Similar results were seen in experimental infections conducted with the same species from Mexico [56]. 

In the present study, the Orange-headed tanager presented the highest amplification capacity (Table 8). However, only two individuals were captured in Boa Viagem, corresponding to less than 1% of all specimens captured. On the other hand, Sayaca tanager, which is another Thraupidae, was the fifth most captured species in Boa Viagem, and among 19 specimens tested at the PLI area, 10.5% were seropositive for WNV (Appendix A). Evidence of WNV exposure has not been commonly reported for these two species, but because its local occurrence and serological evidence of WNV infection, these two species of tanager are potential amplifying hosts of WNV in CE, and other close locations where it occurs. Because of its abundance at the PLI area, and in other parts of the country, Sayaca tanager may be of a particular interest. The potential participation of this species in flavivirus enzootic cycles of transmission in Brazil has already been investigated. Between 2012 and 2013, 29 free-ranging Sayaca tanager individuals collected in the city of São Paulo, southeastern Brazil, tested negative by real-time RT-PCR for flaviviruses [51]. As part of an investigation for WNV in northeast Brazil, four Sayaca tanager individuals were tested, but no evidence of WNV infection was reported [6].

The House sparrow, which is primarily an urban species, is also a potential amplifying host of WNV in Brazil. House sparrows are among the leaders of infected birds in the urban areas worldwide, and their behavior is believed to enhance this species’ involvement, as these birds also nest and roost communally [57]. In experimental infection studies, House sparrows are strong amplifiers of WNV, regardless of the strain of virus evaluated [58]. In the USA, where the dynamics of enzootic transmission of WNV has been more explored, House sparrow and House finch have been shown to be not only exposed countrywide but also are among the most competent amplifying species for WNV [59].

In tropical ecosystems, which present a larger avian species diversity, data regarding the participation of House sparrow and other common species in WNV cycles of transmission remain scarce. In the present study, 3.9% of House sparrows tested were seropositive for WNV, but this species presented the second lowest amplification capacity among species with >5 individuals tested in Boa Viagem (Table 7). However, the House sparrow’s great amplifying potential may not always be predicted by its seroprevalence.

The House sparrow was the most captured species during the investigation (Appendix A), and it is the only one among all species captured in Boa Viagem known to die from infection with the New York 1999 strain of WNV [34], which is the same lineage recently isolated in Brazil [10]. The susceptibility of local sparrows to fatal infection by WNV in Boa Viagem remains unknown, but unnoticed mortality in this species could potentially contribute to a low detection rate for seropositive individuals. Notably, all 14 House sparrows captured and tested from the PLI area were seronegative for WNV. Between 2008 and 2010, eight individuals collected in southern Brazil tested negative for WNV by molecular and serological methods [6]. 

Evidence of WNV circulation in Brazil has been reported since 2011, but potential amplifying bird species in the country remain unknown. In the most recent WNV investigation conducted with free-ranging bird of prey in the country, all samples from southern Brazil tested negative by real-time RT-PCR and PRNT [60]. 

We report here evidence of WNV exposure in different bird species collected in Boa Viagem. As discussed above, it is possible that some of these species that were seropositive for WNV in Boa Viagem may have potentially acted as amplifying hosts in the region. When considering only the species with at least a dozen individuals tested, for a more significant analysis, the seroprevalence observed among them is lower than commonly observed in avian surveys conducted during WNV outbreaks in the USA [17]. The lower prevalence observed in species captured in Boa Viagem may reflect the low or diluted local enzootic circulation of WNV. No other clinical cases of WNV infection have been confirmed in Boa Viagem since the index case was reported in June of 2019.

### 4.4. Arrival of WNV to Boa Viagem

If WNV has silently circulated in birds of Boa Viagem for a while, or if its arrival is a recent event, remains unclear. Between 2012 and 2013, oropharyngeal and cloacal swabs from 529 wild birds from 89 species captured also in the northeast region of the country found no evidence of active circulation of WNV [61]. On the other hand, hemagglutination-inhibition antibodies for WNV have been found in birds sampled after the human case reported in the neighbor state of Piauí (P Lima 2021, personal communication, 4 May). Hypothetically, WNV could have recently circulated in Boa Viagem region in low levels through temporary enzootic cycles of transmission resulting in sporadic spill-over events driven by unknown ecological factors. Ecological factors could include migration movements of potential amplifying birds and/or the increase of vector population density.

This hypothesis is corroborated by the recent report that the ancient strain NY99 was the cause of WNV infection in the equid epizootic occurred in southeast Brazil [13]. The NY99 was the first strain of WNV detected in 1999 in the USA and between 2001 and 2003 went extinct after being displaced by other strains [62]. A low level of transmission potentially due to local ecological conditions could be preventing large outbreaks and evolution of WNV in Brazil, as it has done in the USA. High biodiversity as dilution factor associated with immunity to other flaviviruses and potential limited vector competence in arrival sites could be restricting, at least for now, WNV amplification to isolated and temporary non-continuous cycles of transmission.

The arrival of WNV in Boa Viagem remains unknown, but WNV-infected migratory birds coming from different regions is one of the commonly discussed hypotheses for WNV introductions worldwide [63]. Bird species in the order Charadriiformes, such as shorebirds and terns, are candidates for carrying WNV from North America to South America due to long-lasting high-level viremias, occasional persistent infectious viral loads in skin, and direct, long-distance flights [63]. Another possibility would be the introduction of WNV to CE from other areas of Central and South America, including from other regions of Brazil.

The Eared dove (*Zenaida auriculata*), which is locally called Avoante is commonly seen in flock gatherings during the rainy season, where it forms breeding colonies in the northeast region of Brazil. It has been classified as migratory species and moves in response to rain movements in the Caatinga, where it appears soon after March in southern CE [64,65,66,67]. Eared dove is occasionally hunted in Boa Viagem and the eight year old girl who was seropositive for WNV reported consumption of Eared dove meat between April and September 2019. In a study conducted in Argentina, 157 individuals of Eared dove tested negative for WNV antibodies [50]. In the present study, the only individual that was sampled was seronegative for WNV (Appendix A); therefore, the exposure of Eared dove to WNV in Boa Viagem remains unknown and merits further investigation.

The findings presented here associated with evidence reported in at least ten other states from all regions of the country indicates that WNV is more spread than originally thought [3,4,5,6,7,8,9,10,11,12,68,69,70]. For decades urban areas in Brazil have been heavily hit by Aedini-borne arboviruses, but less impacted by arboviruses transmitted by Culicine mosquitoes. With the recent increase of WNV reports in Brazil, this empty niche in urban and peri-urban areas should be constantly monitored. The large variety of flaviviruses circulating in Brazil, which could generate heterologous antibodies enough to prevent clinical infections by WNV, has been considered one of the main reasons for the absence of large neurological disorder outbreaks caused by WNV in the country. From 513 samples tested in the present study, seven (1.4%) samples were seropositive for SLEV and one (0.2%) for DENV-1. It is noteworthy that these findings are likely to be underestimated since these samples were found at random during differential diagnosis. No specific investigations for SLEV, DENV, and ILHV were conducted in Boa Viagem.

However, additional factors as recent arrival, local ecological characteristics, and vector and vertebrate host competence may also be involved. Heterologous antibodies for other flaviviruses were not enough to prevent the arrival and spread of DENV in the 1980s in a yellow fever endemic state, neither of ZIKV in a dengue-hyperendemic country in 2015. The impact of cross-reactivity observed among flaviviruses also influences serological diagnostic results. The interpretation of serological tests in flavivirus diagnosis is complex and requires careful evaluation [17]. With the establishment of WNV in Brazil, where at least ten other flaviviruses of medical importance circulate, the serological diagnostic of flaviviruses, including WNV, tends to become even more challenging. For that reason, we used a conservative threshold for detection of neutralizing antibodies (90%), and we considered monotypic serologic responses to be the most reliable, as these samples reacted with just one of all viruses employed in the tests, with no indication of crossreaction.

### 4.5. Seroconversion and Potential Maintenance of WNV in Boa Viagem

At least four local domestic birds had four-fold increase in PRNT_90_ titers between September and November of 2019 (Table 5). Moreover, according to the results two more seroconversions occurred among the studied domestic birds. The seroconversions observed in domestic birds reported here are suggestive of recent exposure to WNV. However, stronger evidence relies on a single female helmeted guineafowl that neutralizing antibody titers went from <10 to 160 (Table 5). The seroconversion observed in the other birds would not be enough to confirm the recent circulation of WNV. Some individuals that are sequentially infected by a heterologous flavivirus species can boost antibody levels against the original virus, resulting in a phenomenon known as ‘original antigenic sin’ which can lead to incorrect diagnoses [71]. Samples collected in November of 2019 were not tested for other flaviviruses.

Moreover, except for the index horse case, all vertebrate and mosquito samples that were tested for WNV by real-time RT-PCR were negative. To be more likely detected using direct diagnostic methods in sera of terminal hosts, blood sampling must be performed during the short period of viremia of WNV, which may peak before the onset of neurological clinical signs. A recent study conducted in Brazil, reported genetic evidence of WNV in equine red blood cells by portable nanopore sequencing, suggesting whole blood as a potential alternative sample for the diagnostic of WNV in horses [12]. 

A complementary approach to investigate active infection of WNV would be testing by real-time RT-PCR plasma samples from free-ranging wild birds. As amplifying hosts, passerines present higher-level and longer viremia making their plasma or serum samples more suitable for direct diagnostic methods for WNV. Because of the low volume of blood (<~200 microliters) drawn for most specimens of birds captured in Boa Viagem, the evaluation of WNV exposure in these animals had to be restricted to the investigation of antibodies. In the present study, an additional subset of tissue samples was collected from individuals that unfortunately died during netting and sampling, and will be further investigated. Future investigations would ideally assess not only antibodies, but also viremia and/or antigen of WNV, preferentially but not restricted to the species that were seropositive for WNV in the present study.

### 4.6. Potential Screening Methods for WNV Serological Surveys

Regarding the potential use of HI as a screening method for confirmatory PRNT_90_ for future WNV-specific antibody investigations in Brazil, an optimal screening test should detect all positive samples even at the cost of detecting some false positives. The comparisons between HI and PRNT_90_ results demonstrate that the value of HI as a screening method for PRNT_90_ for WNV was in general limited, varied according to the vertebrate group tested, and was greatly improved by the use of HI B criterion of seropositivity in most groups. While for equid, domestic bird, and small ruminant samples, the efficiency of HI as a screening method was improved by the use of HI B as selection criterion, for human samples, no much difference was observed between the two criteria of selection (Figure 3). These results were below expectations particularly for human samples and indicates that screening HI B criterion should be preferred in future similar WNV surveys in domestic animals.

The use of HI as a screening method demonstrated to be particularly informative to assess the local arbovirus diversity. In the present study, grossly 86% of the human residents of Boa Viagem presented HI titer ≥20 to at least one flavivirus. When the same group of samples was tested for alphaviruses and orthobunyaviruses, 24.3% and 0% were reactive, respectively (Appendix A). Among domestic bird samples, similar results were observed. These HI results demonstrated that local hosts were more exposed to flaviviruses than alphaviruses or orthobunyaviruses combined. Moreover, various samples from domestic birds and humans were seropositive by HI to the alphavirus Mayaro virus, and small ruminants and equids were seropositive to the orthobunyaviruses Maguari, Tacaiuma, and Caraparu viruses. Combining HI and PRNT results, at least eight other arboviruses could be silently circulating in Boa Viagem, and this evidences merit further investigation.

The use of PRNT_90_ also as a preliminary screening method proved to be useful and effective to exclude seronegative samples. With this approach we were able to reduce in more than 70% the number of samples to be confirmed by endpoint PRNT_90_, minimizing the limitation of the use of this assay through adaption for a larger scale use. However, due to the inherent limitations of the PRNT, which includes cell culture production, high level biosafety facilities and highly trained personnel, its use has been restricted to confirmatory tests in reference laboratories. The use of MAC-ELISA as screening method for PRNT could also be an option, but it would need further evaluations. In the present study, from all 12 human samples that were reactive or borderline for WNV by MAC-ELISA, none of them were confirmed to seropositive for WNV by PRNT_90_. (Appendix A). 

Considering the technical limitations of the abovementioned methods, a surrogate screening test for the serological investigation of WNV in Brazil must be continuously pursued. The suitable screening method would be ideally a fast and qualitative kit for the detection of WNV-specific neutralizing antibodies in competitive ELISA or chromatographic immunoassay platforms, as it has been recently and shortly developed worldwide for the diagnostic of severe acute respiratory syndrome coronavirus 2 during the coronavirus disease pandemic. Few options with similar approach are already available for the investigation of flaviviruses including epitope-blocking ELISA and the VecTest-inhibition assay (VIA). Epitope-blocking ELISA has a good sensibility and specificity for animal samples, but it has demonstrated some limitations for human samples [72,73,74]. VIA, which is a field-deployable rapid test has a high sensitivity for a variety of flaviviruses among a population of equines from Brazil and merits further evaluation [75]. 

### 4.7. Environmental Aspects of WNV Circulation in Brazil

WNV has been reported in different environments worldwide. Evidence of WNV circulation has been reported from temperate and low-average temperature countries to warm and tropical areas [1]. Although only with data of countrywide ecological studies, we could better evaluate the influence of climate and ecological conditions for the occurrence of WNV in Brazil, the dry and hot conditions observed in the region of Boa Viagem may have had some influence for the local WNV spillover event. Temperature is being related to rates of WNV transmission, and dry weather has already been related to outbreaks of human WNV infections [76]. High temperature and low rainfall enhanced the probability of chicken seroconversions, which occurred in both urban and rural sites in Guatemala [41]. 

Boa Viagem region has a semiarid hot tropical climate with average annual temperature of 29 °C and average rainfall of 717.7 mm concentrated from January to April. Notably, the first human clinical infection by WNV in Brazil was detected in 2014 in the municipality of Aroeiras do Itaim, which has similar climate conditions and is located in the same biome (*caatinga*) in the neighbor state of Piauí, which is roughly 300 Km distant from Boa Viagem [8]. 

Landscapes where WNV has been detected worldwide vary from wetlands to semiarid regions [77,78]. Similar pattern seems to be occurring in Brazil. Besides the clinical infection of WNV in the state of Piauí [8], the first recognized outbreak of WNV in equines in Brazil occurred at the beginning of the dry season, in an area influenced by the humid Atlantic rainforest biome [9]. Moreover, the first evidences of WNV in Brazil were reported in equids from the Brazilian Pantanal, which is one of world’s largest wetlands with prolonged flooding and dry seasons [3,4,5,6,7]. Drought leads to close contact between avian hosts and mosquitoes around remaining water sources and therefore facilitates the epizootic cycling of WNV within these populations [79]. Because of the reduced flushing of *Culex* mosquitoes during drought events, standing water pools become richer in the organic material that *Culex* spp. need in order to thrive. Such water areas might be attractive for several bird species also, which might increase the bird–mosquito interaction [80,81]. In a study conducted in the USA, a model predicted that without drought nor warmth there would have been 43% fewer cases of WNV in 2018 [82]. Based on climate model predictions for climate change and potentially greater drought occurrence in the future, the frequency and relative risk of WNV outbreaks could increase [76]. This can be particularly important in tropical areas where enzootic transmission cycles have already been identified countrywide, as in Brazil.

## 5. Conclusions

The investigation of WNV in Boa Viagem revealed that not only equines, but also humans and different species of domestic animals and wild birds were locally exposed to WNV. The detection of specific neutralizing antibodies for WNV in certain abundant free-ranging passerine species suggests that birds commonly found in the region may have been involved as amplifying hosts in local transmission cycles of WNV. The increasing reports of WNV circulation throughout Brazil is of medical and veterinarian concern. The detection of a single equine case in absence of human disease in a low populated municipality reveals that the successful communication between the Ministry of Agriculture, Livestock, and Supply and the Ministry of Health is instrumental for a sensitive surveillance system for WNV in Brazil.

## Figures and Tables

**Figure 1 microorganisms-09-01699-f001:**
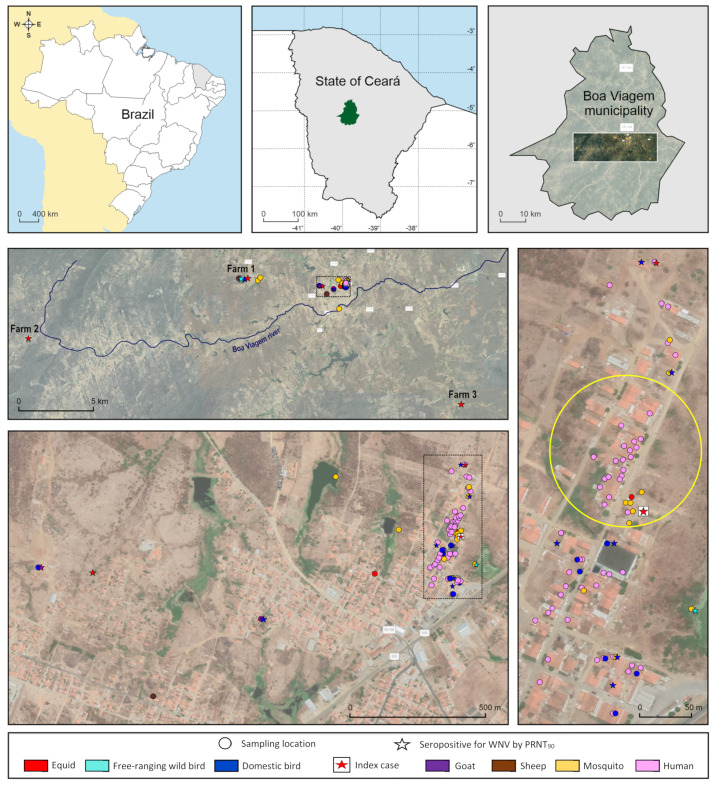
Locations where humans, domestic animals, free-ranging wild birds, and mosquitoes were sampled and tested for WNV in Boa Viagem in 2019. Equids from Farms 2 and 3 had a history of staying at the same property of the index case. Bright yellow circle marks the area where the households of WNV-seropositive humans are located. Seropositive residences are not identified for ethical reasons.

**Figure 2 microorganisms-09-01699-f002:**
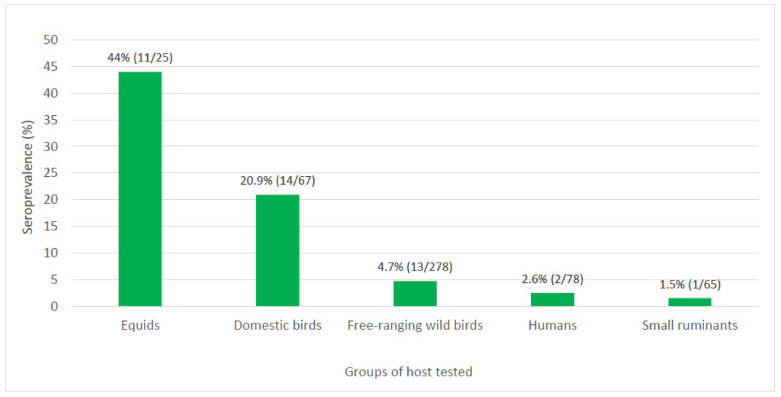
Seroprevalence for WNV by PRNT_90_ in groups of hosts sampled in September 2019 in Boa Viagem, CE.

**Figure 3 microorganisms-09-01699-f003:**
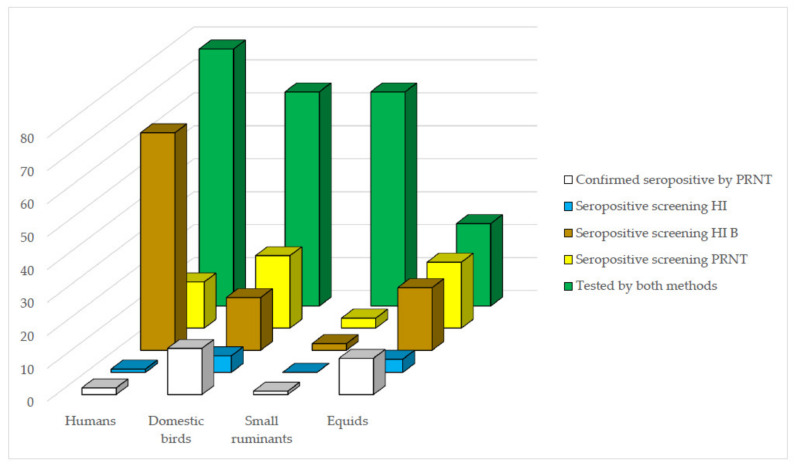
Serum samples of humans and domestic animals from Boa Viagem, CE tested by HI and PRNT_90_.

**Table 1 microorganisms-09-01699-t001:** WNV-seropositive humans sampled in the PLI area in September 2019 in Boa Viagem, CE.

Sample ID	Sampling Date	PRNT_90_ Titer WNV	PRNT_90_ Titer SLEV	PRNT_90_ Titer ILHV	PRNT_90_ Titer DENV-1	PRNT_90_ Titer DENV-2	MAC-ELISA for WNV
PC 033	5 September 2019	320	<10	<10	<10	<10	Non-reactive
PC 024	10 September 2019	≥320	<10	<10	<10	<10	Non-reactive

**Table 2 microorganisms-09-01699-t002:** WNV-seropositive equids by PRNT_90_ in Boa Viagem, CE.

Sample ID	Sampling Date	Site (Location)	PRNT_90_ Titer WNV	PRNT_90_ Titer SLEV	PRNT_90_ Titer ILHV
EQ 002	4 September 2019	PLI area (index case street)	640	80	<10
EQ 003	4 September 2019	PLI area (index case street)	320	<10	<10
EQ 011	7 September 2019	PLI area (neighborhood B)	40	<10	<10
EQ 013	7 September 2019	PLI area (neighborhood B)	320	10	<10
EQ 014	7 September 2019	PLI area (neighborhood B)	40	<10	<10
EQ 020	10 September 2019	Farm 1	160	10	<10
EQ 022	10 September 2019	Farm 1	40	<10	<10
EQ 025	10 September 2019	Farm 1	640	<10	<10
EQ 005	4 September 2019	Farm 2	320	<10	<10
EQ 006	4 September 2019	Farm 2	640	<10	<10
EQ 007	4 September 2019	Farm 3	160	<20	<10

**Table 3 microorganisms-09-01699-t003:** Investigation of seroconversion for WNV in equids in Boa Viagem, CE.

Sample ID	Site (Location)	PRNT_90_ Titer September 2019	PRNT_90_ Titer November 2019
EQ 001	PLI area (index case street)	<10	<10
EQ 002	PLI area (index case street)	640	320
EQ 003	PLI area (index case street)	320	320
EQ 004	PLI area (index case street)	10	10
EQ 011	PLI area (neighborhood B)	40	40
EQ 012	PLI area (neighborhood B)	20	10
EQ 013	PLI area (neighborhood B)	320	320
EQ 015	PLI area (neighborhood B)	<10	10
EQ 019	Farm 1	<10	20
EQ 020	Farm 1	160	320
EQ 021	Farm 1	20	40
EQ 022	Farm 1	40	80
EQ 024	Farm 1	20	<10
EQ 025	Farm 1	640	320
EQ 023	Farm 1	<10	20
EQ 005	Farm 2	320	320
EQ 006	Farm 2	320	320
EQ 007	Farm 3	160	320

**Table 4 microorganisms-09-01699-t004:** Domestic birds that were seropositive for WNV in Boa Viagem, CE.

Sample ID	Sampling Date	Species	Site (Location)	PRNT_90_ Titer WNV	PRNT_90_ Titer SLEV	PRNT_90_ Titer ILHV
GG 027	5 September 2019	*Gallus gallus*	PLI area (index case street)	80	<10	<10
GG 044	5 September 2019	*Gallus gallus*	PLI area (index case street)	320	<10	<10
GG 048	6 September 2019	*Gallus gallus*	PLI area (index case street)	320	<10	<10
GG 008	4 September 2019	*Gallus gallus*	PLI area (index case street)	≥320	20	<20
GG 006	4 September 2019	*Gallus gallus*	PLI area (index case street)	160	<20	<10
GG 030	5 September 2019	*Gallus gallus*	PLI area (index case street)	160	10	<10
GG 047	5 September 2019	? *	PLI area (index case street)	40	<10	<10
GG 051	6 September 2019	*Gallus gallus*	PLI area (index case street)	40	<10	<10
GG 055	6 September 2019	*Gallus gallus*	PLI area (index case street)	40	<10	<10
GG 060	7 September 2019	*Gallus gallus*	PLI area (neighborhood C)	160	<10	<10
GG 056	7 September 2019	*Gallus gallus*	PLI area (neighborhood C)	40	<10	<10
GG 058	7 September 2019	*Gallus gallus*	PLI area (neighborhood C)	40	<10	<10
GG 063	10 September 2019	*Gallus gallus*	Farm 1	40	<10	<10
GG 067	10 September 2019	*Anser anser*	Farm 1	40	<10	<10

* The name of the species was not recorded for sample GG 047.

**Table 5 microorganisms-09-01699-t005:** Seroconversion for WNV in domestic birds in Boa Viagem, CE.

Sample ID	Site (Location)	PRNT_90_ Titer September 2019	PRNT_90_ Titer November 2019
GG 025	PLI area (index case street)	<10	160
GG 034	PLI area (index case street)	10	160
GG 048	PLI area (index case street)	320	2560
GG 051	PLI area (index case street)	40	160

**Table 6 microorganisms-09-01699-t006:** WNV-seropositive free-ranging wild birds in Boa Viagem, CE.

Sample ID	Sampling Date	Site (Location)	Species	PRNT_90_ Titer WNV	PRNT_90_ Titer SLEV	PRNT_90_ Titer ILHV
AS 008	5 September 2019	PLI area (index case street)	*Turdus rufiventris*	80	20	<10
AS 019	5 September 2019	PLI area (index case street)	*Columbina talpacoti*	≥160	40	<10
AS 114	6 September 2019	PLI area (index case street)	*Thraupis sayaca*	80	20	<10
AS 175	6 September 2019	PLI area (index case street)	*Thraupis sayaca*	80	20	<10
AS 219	7 September 2019	PLI area (index case street)	*Forpus xanthopterygius*	320	80	<10
AS 294	9 September 2019	Farm 1	*Passer domesticus*	1280	160	<10
AS 311	9 September 2019	Farm 1	*Coryphospingus pileatus*	640	20	<10
AS 313	9 September 2019	Farm 1	*Thlypopsis sordida*	640	20	<10
AS 353	10 September 2019	Farm 1	*Passer domesticus*	640	40	<10
AS 319	9 September 2019	Farm 1	*Turdus rufiventris*	640	10	<10
AS 341	10 September 2019	Farm 1	*Passer domesticus*	640	40	<10
AS 355	10 September 2019	Farm 1	*Furnarius figulus*	1280	40	<10
AS 367	10 September 2019	Farm 1	*Columbina talpacoti*	320	20	<10

**Table 7 microorganisms-09-01699-t007:** Seroprevalence by PRNT_90_ for WNV among wild bird species with a tested sample size of *n *> 5.

Species Scientific Name	Species Common Name	M	*n*	Prop. (95% CI)
*Turdus rufiventris*	Rufous-bellied thrush	2	6	0.33 (0.09–0.70)
*Columbina talpacoti*	Ruddy ground dove	2	13	0.15 (0.04–0.42)
*Furnarius figulus*	Band-tailed hornero	1	7	0.14 (0.02–0.51)
*Thraupis sayaca*	Sayaca tanager	2	19	0.11 (0.03–0.31)
*Coryphospingus pileatus*	Grey pileated finch	1	11	0.09 (0.02–0.38)
*Passer domesticus*	House sparrow	3	76	0.04 (0.01–0.11)
*Columbina picui*	Picui ground dove	0	35	0.00 (0.00–0.10)
*Volatina jacarina*	Blue-black grassquit	0	16	0.00 (0.00–0.19)
*Columbina minuta*	Plain-breasted ground dove	0	12	0.00 (0.00–0.24)
*Pitangus sulphuratus*	Great kiskadee	0	13	0.00 (0.00–0.23)
*Fluvicola albiventer*	Black-backed water tyrant	0	9	0.00 (0.00–0.30)
*Crotophaga ani*	Smooth-billed ani	0	9	0.00 (0.00–0.30)
*Certhiaxis cinnamomeus*	Yellow-chinned spinetail	0	7	0.00 (0.00–0.35)
*Sporophila albogulari*	White-throated seedeater	0	6	0.00 (0.00–0.39)

*n*—number of individuals tested; M—number of individuals that tested seropositive for WNV.

**Table 8 microorganisms-09-01699-t008:** Amplification of WNV-seropositive free-ranging bird species collected at Boa Viagem, CE.

Species	#Captured	C	#Tested	S	Prop. (95% CI)	A
*Thlypopsis sordida*	2	0.5%	1	100%	1.00 (0.21–1.00)	5000
*Turdus rufiventris*	6	1.6%	6	33.3%	0.33 (0.10–0.70)	1774.2
*Columbina talpacoti*	21	5.5%	13	15.4%	0.15 (0.04–0.42)	1304.4
*Forpus xanthopterygius*	2	0.5%	2	50%	0.50 (0.09–0.91)	1250
*Thraupis sayaca*	20	5.3%	19	10.5%	0.11 (0.03–0.31)	584.3
*Furnarius figulus*	7	1.8%	7	14.3%	0.14 (0.03–0.51)	368.1
*Passer domesticus*	85	22.4%	76	3.9%	0.04 (0.01–0.10)	340.7
*Coryphospingus pileatus*	12	3.2%	11	9.1%	0.09 (0.02–0.38)	265

A (amplification capacity) = C (Relative abundance) × S (seroprevalence)^2^**.**

**Table 9 microorganisms-09-01699-t009:** Adult mosquitoes captured in September 2019 in Boa Viagem, CE, Brazil.

Mosquito Species	Female	Male	Total
*Aedeomyia (Aedeomyia) squamipennis*	6	2	8
*Aedes (Stegomyia.) aegypti*	7	7	14
*Aedes (Georgecraigius) fluviatilis*	1	-	1
*Aedes (Ochlerotatus) scapularis*	2	-	2
*Anopheles (Nyssorhynchus) albitarsis* s.l.	13	8	21
*Anopheles (Nys.) oswaldoi* s.l.	-	1	1
*Anopheles (Nys.) triannulatus* s.l.	11	-	11
*Anopheles (Nys.)* sp.	14	-	14
*Coquillettidia (Rhynchotaenia) nigricans*	10	1	11
*Coquillettidia (Rhy.) venezuelensis*	1	-	1
*Culex (Aedinus) amazonensis*	3	-	3
*Culex (Culex) chidesteri*	5	7	12
*Culex (Cux.) coronator* Complex	15	-	15
*Culex (Cux.) declarator*	38	2	40
*Culex (Cux.) nigripalpus*	18	2	20
*Culex (Cux.) quinquefasciatus*	89	68	157
*Culex (Culex)* spp.	24	-	24
*Culex (Melanoconion) inhibitator*	-	14	14
*Culex (Mel) panacossa*	5	13	18
*Culex (Mel) ribeirensis* *	4	-	4
*Culex (Melanoconion)* sp.	52	4	56
*Culex* spp.	19	2	21
*Mansonia (Mansonia) indubitans*	70	1	71
*Mansonia (Man.) pseudotitillans*	23	-	23
*Mansonia (Man.) titillans*	183	53	236
*Mansonia (Man.)* sp.	35	17	52
*Uranotaenia (Uranotaenia) lowi*	1	-	1
*Uranotaenia* sp.	2	-	2
**TOTAL**	**651**	**202**	**853**

* Females indistinguishable from *Cx. ribeirensis*; the diagnosis could not be confirmed due to the unavailability of male and immature forms.

## Data Availability

Data supporting results can be found at Appendix A.

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
