# Peer review of "West Nile Virus in the State of Ceará, Northeast Brazil"

_microorganisms, 2021, doi:10.3390/microorganisms9081699_

Round 1

Reviewer 1 Report

In the presented manuscript, the authors analyzed the prevalence of West Nile virus in human, animal and mosquito samples from Northeast Brazil.

The manuscript is generally well written. Please see some minor comments below.

General comment

The number of decimal places should be used consistently throughout the manuscript, please check and correct.

Specific comments

Introduction

Lines 76-77: I suggest to replace "terminal hosts" with "incidental or dead-end hosts".

Lines 83-85: please rephrase the sentence "... was reported in 2014, when WNV neutralizing antibodies were detected in a cerebrospinal fluid (CSF) sample from a patient with a flaccid paralysis in the state of ..."

Lines 104-1035: "... samples from humans and animals recovered from arbovirus and neurologic disease surveillances" is not clear, please rephrase.

Materials and Methods

Line 134-135: I suggest to rephrase the sentence "To investigate the role of local mosquito ..."

Line 144: please correct laboratory analysis.

Line 289: Zika virus should be written in an uppercase letter, please correct.

Lines 315-316: human samples were tested for 11 flaviviruses, and domestic animals for 7 flaviviruses - please specify which viruses.

Line 333: please specify which 14 non-flavivirus antigens were tested.

Line 338: please describe briefly MAC- ELISA or add manufacturer (if a commercial test was used).

Results

Tables 3 and 4: What does titer >320 mean? Is it 320 or >320?

Table 6: PRNT titer >160 - please see a comment for tables 3 and 4.

Line 690: please correct 1.3%

Discussion

Line 793: please correct laboratory tests

Line 805: Zika should be written in an uppercase letter, please correct.

Line 1040: please correct 3.9%

Line 1261: please correct 29°C

Line 1273: please correct 25.5°C

Reviewer 2 Report

The manuscript by Flávia Löwen Levy Chalhoub reports the results of an extensive local epidemiological surveillance performed in Brazil after the detection of a WNV-positive horse. The study included several different aspects, from serology to molecular methods, from maintenance and spillover host to vector investigation, from domestic/farm to wild animals. In fact, the study analyzed a high number of sera collected from humans, birds, and equine and ovine animals as well as over 800 mosquitos. The analysis is very thorough and the methods appropriate, especially since several different serological methods were used for testing and positivity criteria were quite stringent. The manuscript is well written, and conclusions are supported by the results.

However, the manuscript is way too long. This is in part due to many redundancies that make the reading tedious and confusing. I think that some sections can be significantly shortened by removing many repetitions and unnecessary details.

M&M: you mention many times in all the sections that blood samples were collected without EDTA, while you can mention it only once in the introductory part and avoid repeating it afterwards. Also lines 145-50 seem superfluous.

Results: you have a long unnecessary introductory part that states the same things outlined in the following paragraphs, for which you also provide figures and tables. I would personally remove entirely lines 381-401 because you say the same things in the following paragraphs and having seroprevalence values for a mixed population that includes humans, horses and goats is pointless anyway. There are also a lot of redundancies between M&M and results (e.g., where samples were collected, like at lines 421-4) that can also be removed.

Discussion: this section is really long (10 pages!) and could also be shortened. Parts that can be shortened are those about which wild birds can be considered amplifying hosts (e.g., lines 995-1014 and 130-137 can be almost entirely removed while all other parts can be significantly shortened), about the screening HI B (lines 1089-1119 have a significant overlap with the results). Also, lines 1151-1163 and 1219-1229 can be removed completely and 1267-1285 shortened (they don’t discuss data of this manuscript and are only tangentially related to the study objective).

Other comments:

- Tables 8-10. Given the high speculative nature of this data, as also highlighted by the authors, I suggest moving these tables to supplementary material. Including these tables in the main text will give to these data the same importance as other data and would imply a non-biased and robust method for evaluating this aspect. For the same reason, I would reduce the text at lines 580-593.

- I would also move Figure 3 to supplementary. Species for which >5 samples were collected are already included in Table 7, while the other species are less relevant.  

- Discussion. To improve readability, you can divide this into subsections.

Lines 315-6. Please indicate which viruses were included in the panel and include testing results in the result section.

Minor:

- Line 55. Did you mean “to the area”?

- Line 104. What do you mean with “humans and animals recovered from arbovirus and neurologic disease surveillances”? It sounds like you are investigating symptomatic subjects. I think this sentence should be phrased more clearly.

- Figure 1. According to journal policies the figure should be located as close as possible to the text where they were first referred to. This figure should therefore be moved to M&M.

- Line 164. It is “exposure”, not “exposition”.

- When you say “domestic birds” I envision perroquets  or canary birds that live in the house. Wouldn’t it be more appropriate in this case to talk about “farmed birds”? Similarly, all wild birds are free-ranging and there is no point in specifying this. I suggest simply using “wild birds”.

- Line 203. “also” and “as well” together is a redundancy. Keep only one.

- Figure 1. It would be clearer if you could add labels for neighborhoods B and C.

- Line 284. I would use “exclude” rather than “reduce” in this sentence.

- Line 329. “named screening HI B” should be included between commas.

- Lines 364-365. This is not acceptable, all methods used should be included in this section.

- Lines 381-90. There is something weird with these numbers. In the first paragraph you identify 58 inconclusives, while in the second you mention only 33+14. Additionally, in the second paragraph I count up to 89 individuals, not 119. Can you please clarify?

- Lines 438-47. Please add how many samples were tested in PCR. Also, Illumina sequencing should be included in M&M.

- Figure 2. It would be informative if you could add here the number of samples tested for each group. This would give a more immediate idea on the robustness of the data for each group.  

- Line 517. What does 57% refer to? Please, specify.

- Line 530. Do you mean at the first timepoint?

- Lines 568-579. This part should be moved to M&M.

- Tables 8-10. In the notes “seroprevalence” is repeated twice.

- I would remove Table 11. A one-line table is pointless.

- Line 657. Did you mean “either in the farm or in the urban area”?

- Lines 793-5. Please, verify the grammar of this sentence.

- Lines 894-8. This sentence is a bit tricky to understand, I suggest rephrasing.

- Line 932. Statistical analysis should be included in M&M

- Lines 938-9. What do you mean with this sentence?

- Lines 944-7. Please, verify the grammar of this sentence.

- Line 1075. What do you mean with “human animal”?

- Line 1195. Please add a reference (the author that gave the personal communication is sufficient)

Round 2

Reviewer 2 Report

Although some redundancies have been removed, the discussion still feels unnecessary long. The paper presents a lot of results, and it is important to provide a cohesive discussion that puts things together, without providing a in depth literature review about every single topic. This is a good paper and I think it is worth at least trying to make the reading more fluid.

Additionally, I still believe that it is pointless to have figures, tables and text stressing the same result and I’d like the authors to consider one more time moving figure 3 and table 8-10 to supplementary material. Maybe you can add an additional column on tables 6 and 7 and add the available A values there, while giving the specifics of the calculations (tables 8-10) as supplementary to avoid repeating species names (which are also already in the text and in Figure 3), N of tested individuals, and prevalence with CI twice.

Minor:

- Lines 318-9. I understand what you say, but the sentence is improperly phrased. Testing for multiple viruses does not reduce/mitigate cross-reactivity (that is the chemical process that happens in the tube) it makes data interpretation less biased. Phrased like this, you make it sound like the more you test the less antibodies cross-react.

- Even if Illumina was just used as a confirmation method, you should still specify how library prep and sequencing were performed since you report results for it.

- Line 1009. I am not sure about the “that”. Maybe a “whose” would be more appropriate?  

Author Response

Although some redundancies have been removed, the discussion still feels unnecessary long. The paper presents a lot of results, and it is important to provide a cohesive discussion that puts things together,without providing a in depth literature review about every single topic. This is a good paper and I think it is worth at least trying to make the reading more fluid.

The careful evaluation of the manuscript conducted by reviewer 2 is greatly appreciated it. All suggestions and comments were carefully read and changes were made throughout the manuscript aiming to address them.

Additionally, I still believe that it is pointless to have figures, tables and text stressing the same result and I’d like the authors to consider one more time moving figure 3 and table 8-10 to supplementary material. Maybe you can add an additional column on tables 6 and 7 and add the available A values there, while giving the specifics of the calculations (tables 8-10) as supplementary to avoid repeating species names (which are also already in the text and in Figure 3), N of tested individuals, and prevalence with CI twice.

Aiming to address reviewer 2’s comments and suggestions, figure 3 and tables 9 and 10 were moved to supplementary material. Table 8 was kept in the manuscript, so the reader can have straight access to both prevalence and amplification tables. The manuscript has now four supplementary tables and one supplementary figure. Additionally, some paragraphs were moved throughout the manuscript aiming to have the topics of Material and methods, results and discussion in similar order of presentation.

Minor:

- Lines 318-9. I understand what you say, but the sentence is improperly phrased. Testing for multiple viruses does not reduce/mitigate cross-reactivity (that is the chemical process that happens in the tube) it makes data interpretation less biased. Phrased like this, you make it sound like the more you test the less antibodies cross-react.

Sentence: “Samples that were reactive for WNV in high PRNT90 titers, as ≥80, were also tested for Saint Louis encephalitis (SLEV) and Ilheus (ILHV) to mitigate cross-reactivity.” was replaced by “Samples that were reactive for WNV in high PRNT90 titers, as ≥80, were also tested for Saint Louis encephalitis (SLEV) and Ilheus (ILHV) viruses as differential diagnosis.”

- Even if Illumina was just used as a confirmation method, you should still specify how library prep and sequencing were performed since you report results for it.

A sub-section was included in M&M to briefly describe the Illumina sequencing

- Line 1009. I am not sure about the “that”. Maybe a “whose” would be more appropriate?

Changed as suggested